# Differential Water Conservation Capacity in Broadleaved and Mixed Forest Restoration in Latosol Soil-Eroded Region, Hainan Province, China

**DOI:** 10.3390/plants13050694

**Published:** 2024-02-29

**Authors:** Suyi Chen, Yanping Huang, Mei Yan, Yujie Han, Kang Wang, Zexian Chen, Dongshuo Ruan, Yan Yu, Zhihua Tu

**Affiliations:** 1School of Tropical Agriculture and Forestry, Hainan University, Haikou 570228, China; 21220954000001@hainanu.edu.cn (S.C.); 22220954000037@hainanu.edu.cn (Y.H.); 20213006716@hainanu.edu.cn (M.Y.); 20196705310020@hainanu.edu.cn (Y.H.); 23220954000037@hainanu.edu.cn (K.W.); 20203106588@hainanu.edu.cn (Z.C.); 20203108246@hainanu.edu.cn (D.R.); 23220954000016@hainanu.edu.cn (Y.Y.); 2Engineering Research Center of Rare and Precious Tree Species in Hainan Province, Haikou 570228, China

**Keywords:** Latosol soil-eroded region, forest restoration, soil layer, litter layer, water conservation capacity

## Abstract

The water conservation capacity of the litter and soil layers of forest ecosystems improves the function of forest ecosystems in conserving soil and water. Plantation restoration plays a key role in preventing soil erosion. In order to evaluate the water conservation capacity of plantation restoration in Latosol soil-eroded region, we analyzed the litter thickness and mass, water absorption process, water holding recovery process, and soil water holding capacity of five restoration types (*Hevea brasiliensis*, *Acacia mangium*, *Eucalyptus robusta*, *Acacia–Eucalyptus*, and *Acacia–Hevea*) in the Mahuangling Watershed, Hainan province. The results showed that the thickness of the litter ranged from approximately 3.42 ± 0.24 to 4.73 ± 0.81 cm, and the litter mass ranged from 5.04 ± 1.52 t·ha^−1^ to 13.16 ± 1.76 t·ha^−1^, with higher litter mass in the SL layer than in the UL layer. The litter mass of *A. mangium* was higher than that of *H. brasiliensis*, *E. robusta*, *Acacia–Eucalyptus*, and *Acacia–Hevea*, which was 3.16 ± 1.76 t·ha^−1^. *A. mangium* forest was significantly higher than other plantation restoration types in terms of the maximum water retention capacity (*Q_max_*) and the effective water retention capacity (*Q_eff_*). The soil bulk weight ranged from approximately 1.52 ± 0.09 to 1.59 ± 0.08 g·cm^−3^, and porosity ranged from 31.77 ± 4.72 to 35.62 ± 3.02%, both of which increased with the depth of the soil layer. The water holding capacity of 0–60 cm soil varied from approximately 12.94 ± 7.91 to 45.02 ± 31.79 t·ha^−1^, with *A. mangium* having the best soil permeability and the strongest soil water holding capacity. The entropy weight method was used to conduct a comprehensive evaluation, and the results showed that the water conservation capacity of the soil layer was 1.26 times higher than that of the litter layer, in which the water conservation capacity of *A. mangium* was the strongest, with a comprehensive evaluation value of 0.2854, which effectively intercepted rainfall and reduced surface runoff. Hence, we suggest that the planting of *A. mangium* should be considered in future ecological restoration projects of the erosion area of Mahuangling in order to improve the function of conserving soil and water in a restoration forest ecosystem.

## 1. Introduction

Soil erosion is a major threat to the global environment [1]; it is one of the environmental issues that requires urgent attention in China [2]. According to the China 2022 Soil and Water Conservation Report, China’s soil erosion area encompasses 265.34 million hectares, of which 41.10% has water erosion. Most of the red soil erosion area in southern China has water erosion [3]. The red soil erosion area is subject to abundant rainfall, diverse forest types, and complex topography; it is one of the five major water erosion types in China, and its water erosion intensity is second only to that of the Loess Plateau of China [4].

Plantation restoration plays an important role in the function of soil and water conservation. The forest canopy layer is able to intercept part of the precipitation that occurs during natural rainfall [5], which is the first layer of the hydrological effect and constitutes the three main functional layers of the vertical structure of the forest, together with the litter layer and the soil layer. The litter layer is characterized by loose [6], porous, and large surface areas [7], which play an important role in preventing surface evaporation, regulating surface runoff, and reducing the damage to the surface induced by rainfall [5]. In addition, the organic matter produced by the decomposition of litter not only effectively improves soil properties and permeability but also provides nutrition for the growth of vegetation [8], which is one of the most important sources of soil nutrients [9]. As the third functional layer, the soil layer is an important link in the process of atmospheric circulation and matter–energy exchange [10]. Rainfall infiltrates down the non-capillary pores of the soil to form free gravity water and subsurface runoff, which can effectively reduce flood peaks and soil loss. The other part is stored in capillary pores for vegetation to absorb for growth and development [11]. Implementing vegetation restoration projects to increase vegetation cover is considered an effective measure to manage soil erosion in watersheds [12]. Many researchers believe [13,14,15] that plantation restoration significantly reduces soil erosion, decreases soil nutrient loss, and improves soil quality. Studies have shown that after vegetation restoration, under the multiple effects of the aboveground forest canopy, the ground litter layer, and underground plant roots, vegetation improved soil structure, significantly enhanced soil erosion resistance, and effectively enhanced the water conservation capacity in the watershed.

The destruction of vegetation is an important factor contributing to regional soil erosion [16]. Through the 1960s to 1980s, the natural forests in the Mahuangling Watershed in Danzhou, Hainan Province, were extensively destroyed, which resulted in soil and water erosion during the rainy season, and the Mahuangling Watershed is now known as a bald mountain with numerous erosion gullies, the deepest of which reaches 30 m. The fragility of the ecological environment in Mahuangling, as a result of anthropogenic disturbances, climate variability, and the destruction of vegetation types, has given rise to a series of ecological and environmental security problems. To control soil erosion and meliorate the ecological system, the Hainan government has implemented large-scale vegetation restoration projects. Starting in 2000, plantation restoration was conducted by planting *A. mangium* and *E. robusta* trees, amongst others. These ecological restoration projects have helped to improve the ecological environment in the Mahuangling Watershed. However, the growth of large areas of *Eucalyptus* forest caused local soil degradation and groundwater depletion [17,18]. As a result, the *Eucalyptus* forest was cut down to plant an *A. mangium* forest instead. In addition, the restoration of vegetation has promoted changes in the structure of vegetation communities [19] and increased biodiversity, which affects the water holding capacity of forests [20,21,22,23,24,25,26]. In order to understand the water holding function after plantation restoration in the eroded area of Latosol, this study explores the differences between the litter and soil hydrological properties of five plantation restoration types in the Mahuangling Watershed, which is of great significance to provide a practical basis for plantation restoration in the Latosol soil-eroded region and improving the ecological environment.

## 2. Results

### 2.1. Litter Thickness and Mass

Forest type has significant effects on total litter thickness (*p* < 0.001). As shown in Table 1, the minimum and maximum litter thicknesses of the different plantation restoration types were 3.42 ± 0.24 cm and 4.73 ± 0.81 cm, respectively, and the order of the litter thicknesses was as follows: *Acacia–Eucalyptus* > *H. brasiliensis* > *Acacia–Hevea* > *E. robusta* > *A. mangium*.

The litter mass under different plantation restoration types is shown in Table 2. The total litter mass differed significantly among five plantation restoration types (*p* < 0.05), and these differences were dependent on the forest. The total litter mass spanned from approximately 5.04 ± 1.52 t·ha^−1^ to 13.16 ± 1.76 t·ha^−1^, where *A. mangium* forest was the highest (13.16 ± 1.76 t·ha^−1^), higher than *Acacia–Eucalyptus* forest (11.94 ± 2.71 t·ha^−1^), *E. robusta* forest (8.87 ± 1.87 t·ha^−1^), *Acacia–Hevea* forest (8.58 ± 1.32 t·ha^−1^), and *H. brasiliensis* (5.04 ± 1.52 t·ha^−1^) (Table 3). There was a difference in the proportion of the litter UL and SL layers in the total litter mass. The range of the change in the proportion of UL to total mass is 24.51% to 36.69%, and that of SL to total mass is 62.56% to 75.49%. The litter mass of five plantation restoration types show that the amount of SL is higher than that of UL.

### 2.2. W_m_, Q_eff_, and Q_max_

The differences in the maximum water holding capacity (*Wm*) of the litter under different plantation restoration types are shown in Figure 1. The *Wm* decreased in the order of *H. brasiliensis* (279.03 ± 25.90%) > *A. mangium* (265.59 ± 18.31%) > *Acacia–Hevea* (259.25 ± 12.51%) > *E. robusta* (257.00 ± 15.04%) > *Acacia–Eucalyptus* (255.20 ± 15.12%) (*p* > 0.05). The *Wm* of the SL layer for *H. brasiliensis* forest was the highest (279.34 ± 35.34%), and that for *Acacia–Eucalyptus* forest was the lowest (255.32 ± 18.95%) among the different plantation restoration types, with a difference of 0.91-fold (*p* > 0.05). The *Wm* in the UL layer was 278.71 ± 22.30% for *H. brasiliensis* forest and was not significantly different from *Acacia–Eucalyptus* forest (255.08 ± 21.87%), *Acacia–Hevea* forest (254.70 ± 20.84%), and *A. mangium* forest (252.42 ± 18.46%). However, all of these were significantly greater than *E. robusta* forest (243.02 ± 17.75%) (*p* < 0.05). The *Wm* of the litter in the SL layer was greater than that in the UL layer under different plantation restoration conditions.

The results show that the litter of *A. mangium* forest (27.05 ± 3.25 t·ha^−1^) had the largest effective water retention capacity (*Qeff*) (Figure 1). This was equal to a 27 mm depth equivalent of rainfall, which was higher than the 22.84 ± 5.17 t·ha^−1^ of *Acacia–Eucalyptus* forest, the 18.65 ± 3.62 t·ha^−1^ of *E. robusta* forest, the 17.41 ± 3.60 t·ha^−1^ of *Acacia–Hevea* forest, and the 10.60 ± 3.96 t·ha^−1^ of *H. brasiliensis* forest (*p* < 0.05). *A. mangium* forest of the SL litter layer could intercept about 15 mm of rainfall, which was larger than the 11.99 ± 3.27 t·ha^−1^ observed for *E. robusta* and significantly higher than *Acacia–Eucalyptus* forest (9.76 ± 8.08 t·ha^−1^), *H. brasiliensis* forest (7.19 ± 3.06 t·ha^−1^), and *Acacia–Hevea* forest (5.05 ± 6.05 t·ha^−1^) (*p* < 0.05). The *Qeff* of the UL layer ranged from 3.41 ± 1.54 t·ha^−1^ to 8.09 ± 2.90 t·ha^−1^ and significantly decreased in the following order: *A. mangium* > *Acacia–Hevea* > *E. robusta* > *Acacia–Eucalyptus* > *H. brasiliensis* (*p* < 0.05). The *Qeff* of the SL layer was significantly higher than that of the UL layer in terms of plantation restoration.

The results show that there were significant differences in the maximum water retention capacity (*Qmax*) under each plantation restoration type (*p* < 0.05) (Figure 1). The *Qmax* of *A. mangium* forest was 32.39 ± 3.80 t·ha^−1^, which is not significantly different from *Acacia–Eucalyptus* forest (27.41 ± 6.16 t·ha^−1^). However, that of *A. mangium* forest was significantly higher than that of *E. robusta* forest (22.14 ± 4.35 t·ha^−1^), *Acacia–Hevea* forest (20.75 ± 4.19 t·ha^−1^), and *H. brasiliensis* forest (12.72 ± 4.63 t·ha^−1^) (*p* < 0.05). The *Qmax* in the SL layer was 22.71 ± 4.39 t·ha^−1^ for *A. mangium* forest. This was 1.11-, 1.59-, and 1.69-fold higher than that of the *Acacia–Eucalyptus*, *E. robusta*, and *Acacia–Hevea* forests, respectively (*p* < 0.05). The *Qmax* of the UL layer ranged from approximately (4.06 ± 1.82 t·ha^−1^) to (9.68 ± 3.39 t·ha^−1^), and its order was as follows: *A. mangium* forest > *E. robusta* forest > *Acacia–Hevea* forest > *Acacia–Eucalyptus* forest > *H. brasiliensis* forest (*p* < 0.05). The *Qmax* of the SL layer was higher than the UL layer *Qmax* among the five plantation restoration types (*p* < 0.05).

### 2.3. Water–Holding Rate of Litter

The water–holding rate of the SL layers and UL layers under different plantation restoration types gradually increases with immersion time, with the most rapid increase occurring within 2 h after the beginning of the experiment. From 2 to 12 h, the water–holding rate slowed, and the change almost stopped at 24 h (Figure 2). The water–holding rate of the SL litter for different plantation restoration types was higher than that of the UL layer under the same immersion time. Within 0.25 h from the start of the experiment, the water–holding rate of the UL litter layers from the different forest types decreased in the following order: *A. mangium* (7.13 ± 1.07 t·ha^−1^) > *Acacia–Hevea* (5.46 ± 0.68 t·ha^−1^) > *E. robusta* (4.99 ± 0.61 t·ha^−1^) > *Acacia–Eucalyptus* (4.56 ± 0.68 t·ha^−1^) > *H. brasiliensis* (3.05 ± 0.36 t·ha^−1^). The water–holding rate of the SL litter layers from the different plantation restoration types decreased in the following order: *A. mangium* (15.63 ± 2.40 t·ha^−1^) > *Acacia–Eucalyptus* (14.64 ± 2.06 t·ha^−1^) > *E. robusta* (9.93 ± 1.31 t·ha^−1^) > *Acacia–Hevea* (9.90 ± 1.17 t·ha^−1^) > *H. brasiliensis* (6.96 ± 1.19 t·ha^−1^). The water–holding rate and immersion time of the litter in the different plantation restoration types showed a logarithmic function relationship.

### 2.4. Water Absorption Rate of Litter

The water absorption rates of the UL and SL litter layer were highest at the start of the experiment and decreased rapidly in the first 2 h. The rate gradually slowed down after 2–12 h and was nearly unchanged after 12 h (Figure 3). The water absorption rate of the SL layer was relatively larger than that of the UL layer at the same time of water immersion. In the first 15 min of the experiment, the water absorption rates of the UL layer of *A. mangium*, *Acacia–Hevea*, *E. robusta*, *Acacia–Eucalyptus,* and *H. brasiliensis* reached 28.53 ± 4.29 t·ha^−1^·h^−1^, 21.85 ± 2.72 t·ha^−1^·h^−1^, 19.96 ± 2.43 t·ha^−1^·h^−1^, 18.26 ± 2.74 t·ha^−1^·h^−1^, and 12.20 ± 1.45 t·ha^−1^·h^−1^, respectively, and the water absorption rates of the SL layer of *A. mangium*, *Acacia–Eucalyptus*, *E. robusta*, *Acacia–Hevea*, and *H. brasiliensis* reached 62.50 ± 9.61 t·ha^−1^·h^−1^, 58.56 ± 8.24 t·ha^−1^·h^−1^, 39.72 ± 5.22 t·ha^−1^·h^−1^, 39.58 ± 4.70 t·ha^−1^·h^−1^, and 27.82 ± 4.75 t·ha^−1^·h^−1^, respectively. An exponential relationship was fitted between the water absorption rate and the water loss time for both litter types of the five plantation restoration types.

### 2.5. Recovery Characteristics of Water Holding Capacity of Litter

The cumulative water–loss ratio of the UL and SL layers from the different plantation restoration types did increase substantially with an increase in the water loss time within 0–4 h (Figure 4). However, the cumulative water–loss ratio slowly increased after 4 h of water loss time, and the UL and SL layers in the *A. mangium* forest were larger than those in the other plantation restoration types. Given the same water loss time, the cumulative water–loss ratio of the SL layer is higher than that of the UL layer. After 72 h of water loss time, the cumulative water–loss ratio of the UL and SL layers reached 11.10 ± 0.75 t·ha^−1^ and 24.71 ± 2.66 t·ha^−1^, respectively, for the *A. mangium* forest, 9.86 ± 0.81 t·ha^−1^ and 15.78 ± 1.41 t·ha^−1^ for the *E. robusta* forest, 8.93 ± 0.74 t·ha^−1^ and 15.14 ± 1.70 t·ha^−1^ for the *Acacia–Hevea* forest, 7.91 ± 0.67 t·ha^−1^ and 24.17 ± 2.41 t·ha^−1^ for the *Acacia–Eucalyptus* forest, and 4.72 ± 0.46 t·ha^−1^ and 11.17 ± 1.33 t·ha^−1^ for the *H. brasiliensis* forest. The relationship between the cumulative water–loss ratio and the water loss time was fitted to a logarithm model observed in the five plantation restoration types.

In the five plantation restoration types, the litter water loss rate of the UL and SL layers followed the same trend (Figure 5), which were highest within 0.25 h after the experiment began, and the rates declined quickly within 2 h. From 2 to 12 h, the litter water loss rates slowed, and then they were nearly unchanged after 12 h (Figure 5). Within 0.25 h from the start of the experiment, the water loss rates of the UL litter layers among the different plantation restoration types were as follows: *A. mangium* (38.55 ± 4.25 t·ha^−1^·h^−1^) > *E. robusta* (34.60 ± 2.57 t·ha^−1^·h^−1^) > *Acacia–Hevea* (31.78 ± 3.90 t·ha^−1^·h^−1^) > *Acacia–Eucalyptus* (27.96 ± 3.28 t·ha^−1^·h^−1^) > *H. brasiliensis* (15.21 ± 1.49 t·ha^−1^·h^−1^). The water loss rates of the SL litter layers among the different plantation restoration types were as follows: *A. mangium* (84.40 ± 14.31 t·ha^−1^·h^−1^) > *Acacia–Eucalyptus* (80.12 ± 8.31 t·ha^−1^·h^−1^) > *E. robusta* (58.50 ± 5.07 t·ha^−1^·h^−1^) > *Acacia–Hevea* (52.59 ± 6.46 t·ha^−1^·h^−1^) > *H. brasiliensis* (35.33 ± 6.43 t·ha^−1^·h^−1^). There was a significant exponential relationship between the water loss rate and the water loss time of the litter.

### 2.6. Soil Water Holding Capacity

As shown in Table 3, the bulk density varied among the five plantation restoration types. The bulk density at depths of 0–10 cm under different plantation restoration types was significantly lower than that of other soil depths (*p* < 0.05). At 0–60 cm soil depth, the bulk density of different plantation restoration types was in the following order: *E. robusta* (1.59 ± 0.08 g·cm^−3^) > *A. mangium* (1.55 ± 0.11 g·cm^−3^) > *H. brasiliensis* (1.53 ± 0.08 g·cm^−3^) > *Acacia–Eucalyptus* (1.52 ± 0.09 g·cm^−3^) = *Acacia–Hevea* (1.52 ± 0.09 g·cm^−3^).

Total porosity (0–60 cm) under various plantation restoration types is outlined in Table 3. Soil total porosity was 35.62 ± 3.02%, 34.87 ± 3.27%, 34.45 ± 4.29%, 32.31 ± 4.91%, and 31.77 ± 4.72% for *Acacia–Hevea* forest, *H. brasiliensis* forest, *Acacia–Eucalyptus* forest, *A. mangium* forest, and *E. robusta* forest, respectively (*p* > 0.05). For the 0 to 60 cm soil layer, soil capillary porosity was highest in *Acacia–Hevea* forest (33.95 ± 2.81%), followed by *Acacia–Eucalyptus* forest and *H. brasiliensis* forest, and then *A. mangium* forest. *E. robusta* forest had the lowest value (29.91 ± 4.99%) (*p* > 0.05). Soil non-capillary porosity was 2.03 ± 1.33%, 1.86 ± 2.48%, 1.67 ± 1.25%, 1.56 ± 0.96%, and 1.55 ± 0.89% for *H. brasiliensis* forest, *E. robusta* forest, *Acacia–Hevea* forest, *Acacia–Eucalyptus* forest, and *A. mangium* forest, respectively (*p* < 0.05).

There was a significant difference in the water holding capacity between different soil layers among the five plantation restoration ecosystems (*p* < 0.05). Table 3 shows that the soil of the *A. mangium* forest type increased with soil depth, while the opposite trend was the case for the other forest types. Overall, the soil water storage of *A. mangium* displayed higher values, while the minimum value of soil water storage occurred in *E. robusta*. The order of the soil water holding capacity at all depths was *A. mangium* (45.02 ± 31.79 t·ha^−1^) > *H. brasiliensis* (20.25 ± 13.28 t·ha^−1^) > *Acacia–Hevea* (16.71 ± 12.49 t·ha^−1^) > *Acacia–Eucalyptus* (15.90 ± 9.49 t·ha^−1^) > *E. robusta* (12.94 ± 7.91 t·ha^−1^) (*p* < 0.05) (Table 3). Specifically, the median values of the water holding capacity at 0–60 cm in *A. mangium*, *H. brasiliensis*, *Acacia–Eucalyptus*, *Acacia–Hevea,* and *E. robusta* were 34.80 t·ha^−1^, 15.25 t·ha^−1^, 13.60 t·ha^−1^, 10.30 t·ha^−1^, and 9.80 t·ha^−1^, respectively (Figure 6).

### 2.7. Comprehensive Evaluation of Water Conservation Capacity

In order to evaluate the hydrological effects of the soil and litter layers under different plantation restoration types, the factors of the soil layer and litter layer were quantified using the entropy weight method in order to make a comparison. The litter layer factors include litter thickness, litter mass, *W_m_*, *Q_eff_*, and *Q_max_*. Soil layer includes soil bulk density, non-capillary porosity, capillary porosity, total porosity, and soil field capacity.

The order for weight is as follows: soil water holding capacity > non-capillary porosity > *W_m_* > litter thickness > total porosity > capillary porosity > *Q_max_* > total litter mass > *Q_eff_* > soil bulk density (Table 4). The comprehensive evaluation value of the soil layer was 1.26-fold higher than that of the litter layer. In general, the water conservation capacity of *A. mangium* was the largest, with a comprehensive evaluation value of 0.2854, followed by *E. robusta* (0.2500), *Acacia–Hevea* (0.1900), *Acacia–Eucalyptus* (0.1740), and *Hevea brasiliensis*, which had the smallest water conservation capacities (0.1006) (Table 5).

## 3. Discussion

This study focuses on the hydrological properties of different plantation restoration types from the perspective of litter and soil layers. The litter layer is the “umbrella” of the soil layer. Several research studies have indicated that a larger litter thickness and mass prevents rain from damaging the soil structure and effectively improves the water conservation capacity of forests [27,28,29]. There are differences in the thickness and mass of litter in different plantation restoration types, which is an important factor for the variation in the hydrological characteristics. In this study, the litter thickness and litter mass of five plantation restoration types differed significantly (*p* < 0.05). The thickness of the litter layer ranged from approximately 3.42 to 4.73 cm, and the litter mass ranged from approximately 5.04 to 13.16 t·ha^−1^. The thickness of the litter layer of *Acacia–Eucalyptus* forest was the largest, which was conducive to reducing the evaporation of surface soil. In addition, the SL layer of the litter mass of *Acacia–Eucalyptus* forest accounted for a large proportion of the total mass, indicating that the litter decomposition rate was faster. The litter mass of *A. mangium* was larger than that of the other four plantation restoration types, implying that *A. mangium* had the greatest interception capacity. The mass of the litter in the SL layer is greater than that in the UL layer, which is consistent with the results of Lan et al.’s study [30]. The reason for this is that the decomposition rate of the litter is faster, and the density of the litter after decomposition is larger [31]. Moreover, in previous years, the existing mass of litter was affected by the amount of forest litter [32], meaning that it is necessary to monitor the annual dynamic change in litter in order to further understand the hydrological effect of litter.

The *W_m_* was influenced by the composition, structure [33], and rate of decomposition [34] of the litter. The *W_m_* varies from (255.20 ± 15.12%) to (279.03 ± 25.90%), implying that the litter absorbs about 2.55 to 2.79 times its own dry weight of water. *Acacia–Eucalyptus* showed the smallest *W_m_*, and *H. brasiliensis* showed the largest. The *W_m_* of the SL layer of the litter was greater than that of the UL layer, which is in agreement with the results of Zhao et al.’s study [23]. The reason for this is that the litter SL layer had large pores and was looser, and it is also related to the fact that the litter mass was larger in the SL layer than in the UL layer [35,36].

Litter interception is an important part of the forest hydrological regulation process [37,38]. The *Q_eff_* was influenced by rainfall intensity, rainfall duration, and the amount of litter [39]. In this study, there was a significant difference (*p* < 0.05) in the *Q_eff_* of litter among different plantation restoration types. We found that *A. mangium* had the highest effective interception of 27.06 mm. Tu et al. [40] confirmed that this was related to the leaf areas of *A. mangium*, *Acacia–Eucalyptus*, *E. robusta*, *Acacia–Hevea*, and *H. brasiliensis*, which were 22.84 mm, 18.65 mm, 17.41 mm, and 10.60 mm, respectively. The results showed that *A. mangium* has a high retention capacity, which reduces surface runoff and contributes to the improvement in soil and water conservation.

The water holding processes of litter reflect its hydrological properties [41]. The water absorption rates of the litter increased rapidly at the beginning of the experiment due to the difference in the water potential. In this study, the water absorption rate in the first 2 h changed the most, where it reduced from 12.13~62.50 t·ha^−1^·h^−1^ to 1.80~10.27 t·ha^−1^·h^−1^, and the water–holding ratio of the litter ranged from 3.04~15.63 t·ha^−1^ to 3.60~20.54 t·ha^−1^. Leaf area is an important factor affecting the water holding capacity of litter [42]. As shown in the water holding process of litter, the changes in the water–holding ratio and water absorption rates of the litter were most significant in *A. mangium* forest. The reason for this is that under the same mass conditions, the leaf surface area of *A. mangium* in contact with water was larger than that of *E. robusta* and *H. brasiliensis* [43,44].

The cumulative water–loss ratio and the water loss rate of litter reflect the water retention capacity of the litter, which is one of the important indexes for determining the water conservation capacity of litter. In this study, the water loss rate of the different vegetation restoration types was significantly changed before 2 h, gradually smoothed out after 2 h, and basically stopped at 72 h. This is attributed to the large water content of the litter in the early stage of water loss, where gravity caused part of the water to infiltrate into the soil layer, and a small part of the water evaporated. This is consistent with the results of Dong et al.’s study [45]. We observed the water loss process and found that the cumulative water–loss ratio of *H. brasiliensis* was the smallest, and the cumulative water–loss ratio of *A. mangium* was the largest. This is because the cumulative water–loss ratio was related to the *R_max_*, implying that the greater the *R_max_* of the litter, the more water the litter loses under certain conditions. This is consistent with the results of Du et al.’s study [46], who focused on the recovery characteristics of the water holding capacity of the litter in Xiaowutai Mountain. In order to better understand the water conservation capacity of the litter, it is also necessary to study the influence of litter characteristics on the recovery process of the water holding capacity.

Intercepting the soil layer is a very important process in forest areas, accounting for about 85% of forest water conservation and directly affecting surface runoff [47,48]. This study showed that the water holding capacity of the soil layer was 1.26-fold more than that of the litter layer. Soil physical properties are important factors to consider in the plantation restoration process as they influence the infiltration process of the soil layer in different plantation restoration types [49]. Soil bulk and porosity are influenced by the state of soil development, which results in differences in soil physical properties [50]. A high soil capacity indicates a compact soil lacking in aggregate structure, while the opposite is true for a loose and porous soil [6,7]. The greater the capillary porosity, the greater the proportion of the effective water used for vegetative growth and development, and the more favorable it is for plant root absorption [9]. The capillary porosity, non-capillary porosity, and total porosity showed an increasing trend with decreasing soil bulk in this study. The soil water holding capacity is considered to be an important indicator for evaluating soil hydrological properties [10,49,50]. We found that the soil water holding capacity of *A. mangium* was the best compared to the other plantation restoration types. Further studies need to be conducted to study the moisture dynamics of the soil in the erosion area and to understand the hydrological properties of the soil layers.

We discovered that *A. mangium* forest had a better water conservation capacity than the other restoration types, as evidenced by the higher water holding and interception capacity of the litter layer and the water holding capacity of the soil layer, allowing *A. mangium* forests to perform better water conservation functions during high-intensity rainfall. In conclusion, the eroded area of Latosol of the Mahuangling Watershed, Danzhou, Hainan, China, was suitable for planting *A. mangium* forest for soil and water conservation. 

## 4. Materials and Methods

### 4.1. Study Sites

This study was carried out in the Mahuangling Watershed, a Monitoring Station of the Soil and Water Conservation (109°28′12.79″ E~109°28′26.04″, 19°45′4.51″~19°45′23.53″ located in Danzhou City, Hainan Province, China). The region is a typical eroded area of Latosol, which is one of the serious soil erosion areas in Hainan Province. The watershed has a tropical monsoon climate, with a mean annual precipitation of 1815 mm, and 80~85% of the total precipitation occurs as rainfall between May and October. The annual average temperature is 23.5 °C. The unreasonable cutting down of vegetation by local residents is an important cause of soil erosion and ecological environment deterioration in the Mahuangling Watershed. In the 2000s, large-scale plantation restoration projects began to be implemented to improve the ecological environment, including the plantation of *A. mangium* forest, *E. robusta* forest, *H. brasiliensis* forest, *Acacia–Eucalyptus* forest (mix of *A. mangium* and *E. robusta*), *Acacia–Hevea* forest (mix of *A. mangium* and *H. brasiliensis*) (Figure 7). With the implementation of ecological projects, soil and water loss in the region has been controlled and alleviated, and the forest coverage rate has reached above 75% [20]. These sites also host species such as *Cyperus rotundus*, *Oplismenus undulatifolius*, *Arthraxon hispidus*, *Embelia laeta*, *Schima superba*, and *Litsea glutinosa*.

### 4.2. Collection of Litter and Soil Samples

The typical and representative vegetation restoration types we selected were *A. mangium* forest, *E. robusta* forest, *H. brasiliensis* forest, *Acacia–Eucalyptus* forest (*A. mangium* and *E. robusta*), and *Acacia–Hevea* forest (*A. mangium* and *H. brasiliensis*), all of which are evergreen tree species. In July 2022, three 20 m × 20 m quadrats were established (the plot that was selected did not cross the erosion trench and was at least 5 m away) in each vegetation restoration type to investigate the basic status of the vegetation (Table 6). Furthermore, three 0.50 m × 0.50 m litter quadrats were randomly selected. A total of 90 bags of litter in the un-decomposed layer and semi-decomposed layer were collected from each litter sample. Three soil profiles were excavated in the standard sample plot, and a cutting ring (100 cm^3^) was used to sample the 0–10, 10–20, 20–40, and 40–60 cm soil layers; a total of 360 samples were collected (each layer with two cutting rings).

### 4.3. Laboratory Analyses

The litter mass was measured using the drying method outlined in our previous study [21]. The litter samples were dried in an oven at 75 °C, and the dry weight of the litter was recorded to determine the accumulation of litter. The indoor water soaking method was used to observe the water holding process of the litter [22]. The litter samples were weighed (5 g) into nylon mesh bags soaked in water. The weight of the nylon bag was measured at 0.25, 0.5, 1, 2, 4, 6, 8, 12, and 24 h of immersion. After 24 h, the nylon bag was placed on the sand, and its weight was measured at 0.25, 0.5, 1, 2, 4, 6, 12, 24, 36, 48, and 72 h. The formula for the indicators of the litter is as follows [23]:*W_m_* = (*W*_24h_ − *G*_0_)/*G*_0_ × 100(1)
where *W_m_* is the maximum water holding capacity(%); *W*_24h_ is the litter water content when soaking for 24 h, which is the maximum water content of the litter (g); and *G* and *G*_0_ are the litter fresh weight and litter drying weight.
*W*_0_ = (*G* − *G*_0_)/*G*_0_ × 100(2)
*Q_max_* = (*W_m_
*− *W*_0_) × M(3)
*Q_eff_* = (0.85 × *W_m_ *− *W*_0_) × M(4)
where *W*_0_ is the natural water content (%), M is the litter mass (t·ha^−1^), *Q_max_* is the maximum water retention capacity (t·ha^−1^), and *Q_eff_* is the effective water retention capacity (t·ha^−1^).

In the laboratory, the cutting ring method was used to determine the soil physical properties and soil water holding capacity. The specific experimental steps and calculation methods were carried out following the methodology for the determination of forest soil moisture and physical properties according to the Forestry Industry Standard of the People’s Republic of China (LY/T 1215-1999) [24]:*S* = 10,000 × *hp*(5)
where *S* is the water holding capacity (t·ha^−1^), *h* is the depth of the soil layer (m), and *p* is the non-capillary porosity (%).

### 4.4. Comprehensive Evaluation of Water Conservation Capacity

To evaluate the water conservation capacity of the litter layer and soil layer under different restoration types, the entropy weight method was used to quantify each index. The calculation formula is as follows [25,26]:

(1) The entropy value of each indicator was calculated based on the following formula, denoted as *F_i_*:(6)$$F_{i}=-k\sum_{j}^{m}W_{ij}\ln W_{ij}\;\left(i=1,\;\cdots ,\;n;\;j=1,\;\cdots ,\;m\right)\;$$
(7)$$W_{ij}=\frac{r_{ij}}{\sum_{i=1}^{m}r_{ij}\;\;}$$
where *F_i_* is the entropy value; *k* is a constant, $\frac{1}{\ln m}$. *W_ij_* is the contribution of the *i*-th vegetation restoration type *F_i_* under the *j*-th evaluation index, assuming *W_ij_* = 0, and then *W_ij_*In*W_ij_* = 0. *I* is defined as the number of vegetation restoration patterns. *J* is defined as the number of evaluation indexes.

(2) The weight of each indicator was calculated, defined as *Z_i_*, based on the following formula:(8)$$Z_{i}=\frac{1-F_{i}}{m-\sum_{i=1}^{n}F_{i}}\;\left(i=1,\;2,\;\cdots ,n\right)$$
where *Z_i_* is the weight, *F_i_* is the entropy value, and *i* is the vegetation restoration type.

### 4.5. Statistical Analysis

SPSS 26 and Microsoft Excel 2021 were used to analyze the data. The significance of the differences in the hydrological properties between the litter and soil layers was compared using the least significant difference (LSD) method and a one-way ANOVA, and data are expressed as mean ± standard error (Mean ± SE). The significance level was set at *p* < 0.05. The graphs and tables shown in this paper were produced using Origin 2023.

## 5. Conclusions

Different plantation restoration types significantly change the hydrological characteristics of soil and litter layers, thus affecting the water holding capacity of the ecosystem. Here, we studied the hydrological effects of the litter layer and soil layer on different plantation restoration types in the Mahuangling Watershed. We found that the water holding capacity of the litter SL layer was greater than that of the UL layer for different plantation restoration types. Also, the soil layer has a low bulk density and high porosity, meaning that it has better aeration and permeability too. In addition, the water holding capacity of the soil layer was 1.26 times higher than that of the litter layer. Notably, the litter mass, water absorption rates, water holding capacity, and water holding restoration capacity in the litter layer of *A. mangium* forest were better than those of the other plantation restoration types, while the water holding capacity of the soil layer of *A. mangium* was the highest, which demonstrated greater potential for improving the water holding capacity of the Mahuangling Watershed. In summary, we recommend that for future plantations in forests, *A. mangium* forest is chosen to promote the ecological environment of Mahuangling.

## Figures and Tables

**Figure 1 plants-13-00694-f001:**
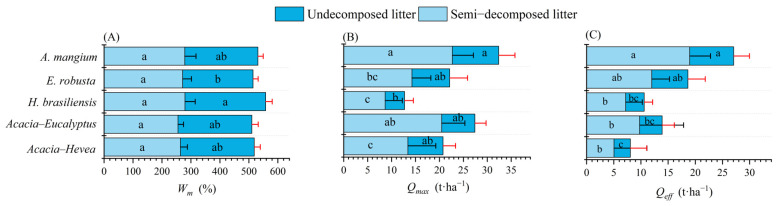
The *Wm*, *Qmax*, and *Qeff* in the undecomposed litter (UL) layer and semi-decomposed litter (SL) layer from the five plantation restoration types. (**A**) The maximum water holding capacity (*Wm*); (**B**) the maximum water retention capacity (*Qmax*); (**C**) the effective water retention capacity (*Qeff*). Different lowercase letters indicate significant differences (*p* < 0.05).

**Figure 2 plants-13-00694-f002:**
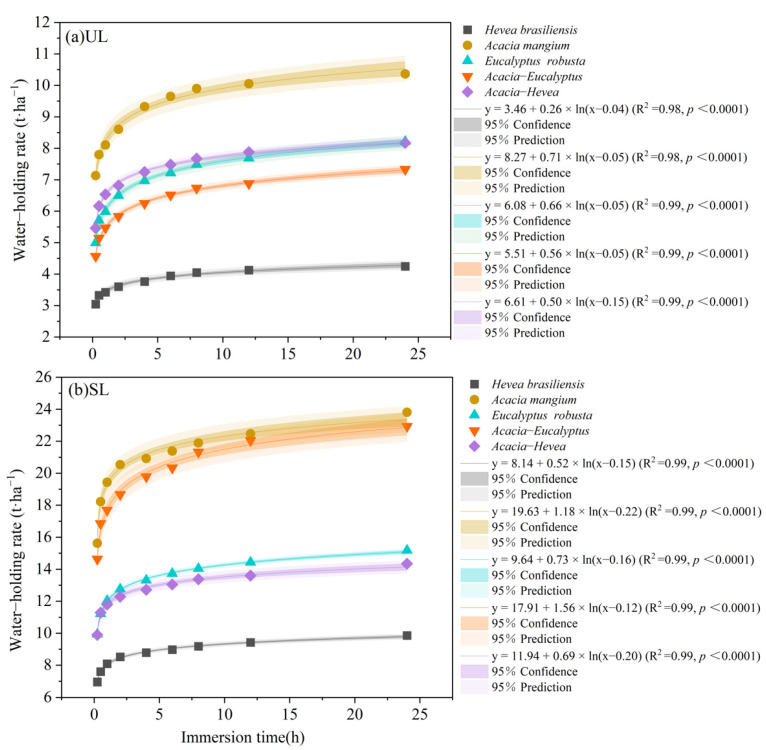
Comparison of water–holding rate of litter in different plantation restoration types: (**a**) un-decomposed layer (UL) and (**b**) semi-decomposed layer (SL) of *H. brasiliensis*, *A. mangium*, *E. robusta*, *Acacia–Eucalyptus*, and *Acacia–Hevea*.

**Figure 3 plants-13-00694-f003:**
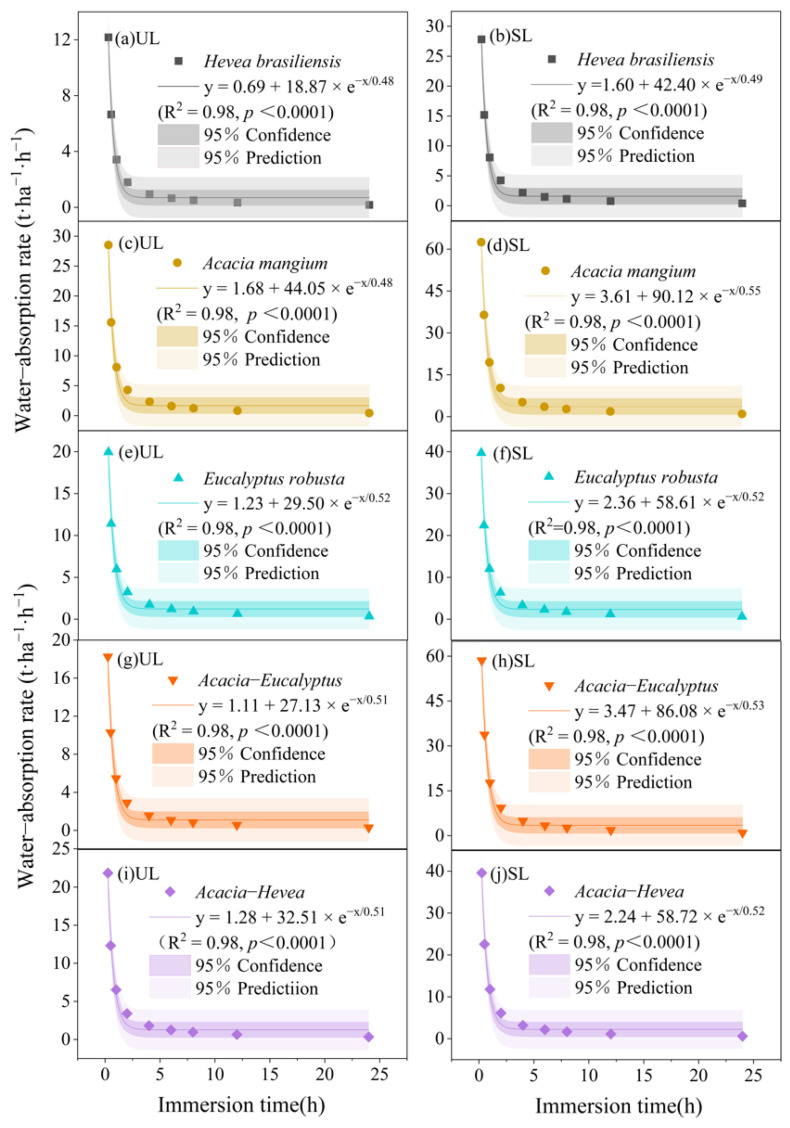
Comparison of water absorption rate of litter in different plantation restorations of un-decomposed litter (UL) layer for (**a**) *H. brasiliensis*, (**c**) *A. mangium*, (**e**) *E. robusta*, (**g**) *Acacia–Eucalyptus*, and (**i**) *Acacia–Hevea* and semi-decomposed litter (SL) layer for (**b**) *H. brasiliensis*, (**d**) *A. mangium*, (**f**) *E. robusta*, (**h**) *Acacia–Eucalyptus*, and (**j**) *Acacia–Hevea*.

**Figure 4 plants-13-00694-f004:**
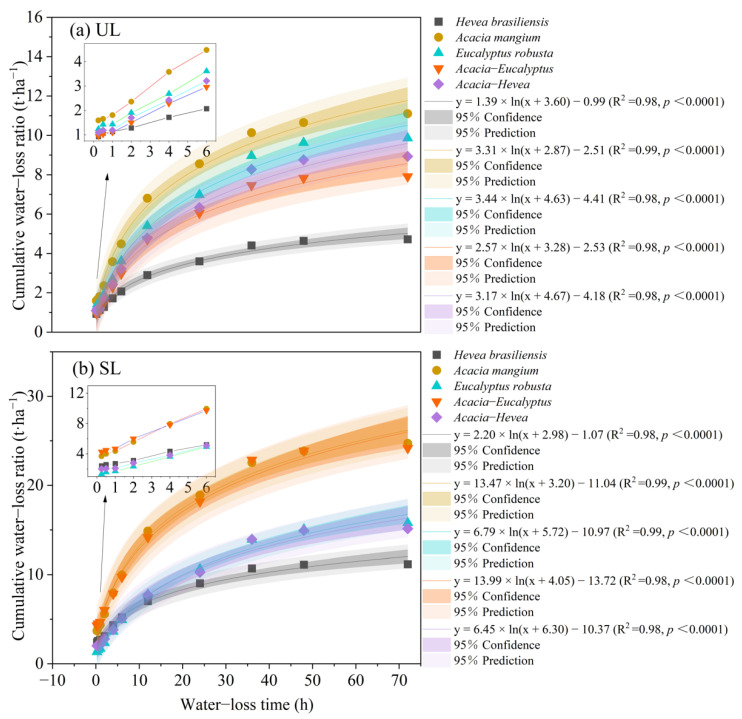
Comparison of cumulative water–loss ratio of litter in different plantation restoration types: (**a**) un-decomposed layer (UL) and (**b**) semi-decomposed layer (SL) of *H. brasiliensis*, *A. mangium*, *E. robusta*, *Acacia–Eucalyptus*, and *Acacia–Hevea*.

**Figure 5 plants-13-00694-f005:**
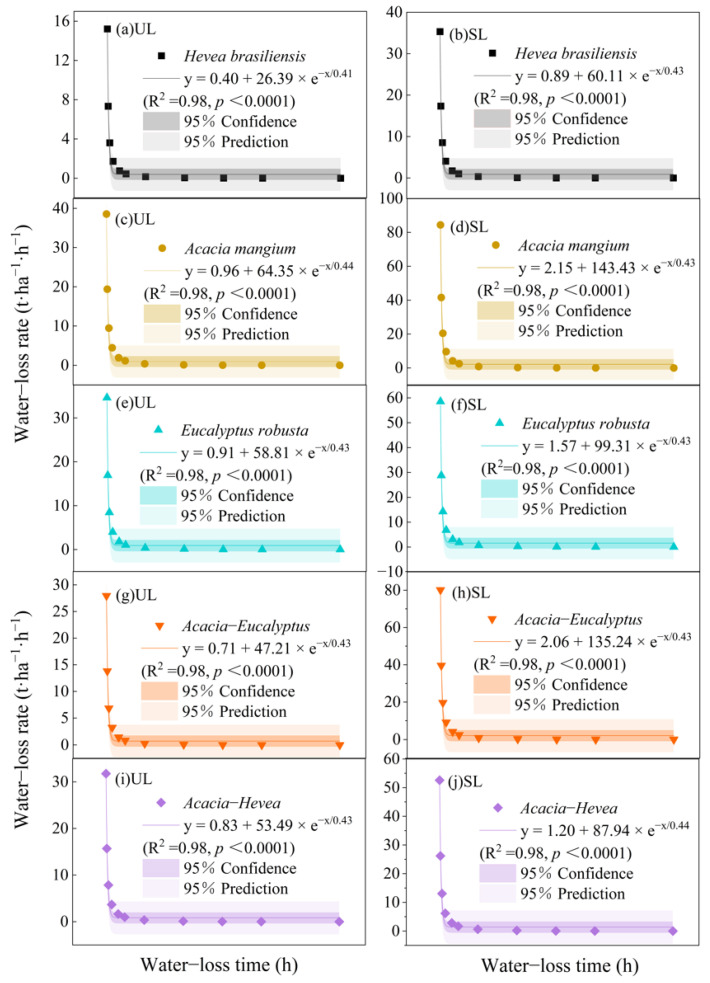
Comparison of water loss rate of litter in different plantation restoration types of un-decomposed litter (UL) layer for (**a**) *H. brasiliensis*, (**c**) *A. mangium*, (**e**) *E. robusta*, (**g**) *Acacia–Eucalyptus*, and (**i**) *Acacia–Hevea* and semi-decomposed litter (SL) layer for (**b**) *H. brasiliensis*, (**d**) *A. mangium*, (**f**) *E. robusta*, (**h**) *Acacia–Eucalyptus*, and (**j**) *Acacia–Hevea*.

**Figure 6 plants-13-00694-f006:**
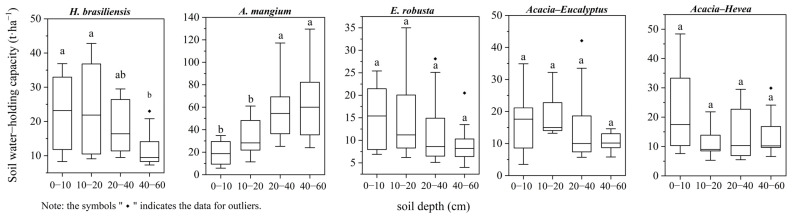
Soil water holding capacity at depths of 0–10, 10–20, 20–40, and 40–60 cm for *A. mangium*, *E. robusta*, *H. brasiliensis*, *Acacia–Eucalyptus,* and *Acacia–Hevea*. Different lowercase letters indicate significant differences (*p* < 0.05).

**Figure 7 plants-13-00694-f007:**
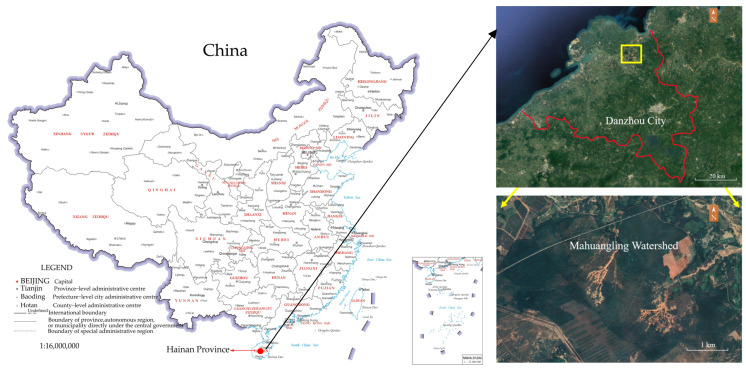
Locations and photos of the sampling sites at the Mahuangling Soil and Water Conservation Monitoring Station.

**Table 1 plants-13-00694-t001:** The litter thickness and litter mass in the un-decomposed litter (UL) layer and semi-decomposed litter (SL) layer of the forest types of *A. mangium*, *E. robusta*, *H. brasiliensis*, *Acacia–Eucalyptus,* and *Acacia–Hevea*. For each category, different lowercase letters indicate a significant difference from five plantation restoration types (*p* < 0.05).

Type of Community	Litter Thickness/cm	Total Litter Mass/(t·ha^−1^)	Un-Decomposed Layer		Semi-Decomposed Layer	
			Litter Mass/(t·ha^−1^)	Proportion/%	Litter Mass/(t·ha^−1^)	Proportion/%
*H. brasiliensis*	4.36 ± 0.77 a	5.04 ± 1.52 c	1.54 ± 0.58 b	30.46	3.51 ± 1.13 b	69.54
*A. mangium*	3.42 ± 0.24 b	13.16 ± 1.76 a	4.16 ± 1.08 a	31.64	8.99 ± 1.09 a	68.36
*E. robusta*	3.57 ± 0.56 b	8.87 ± 1.87 b	3.32 ± 1.22 a	37.44	5.55 ± 1.53 b	62.56
*Acacia–Eucalyptus*	4.73 ± 0.81 a	11.94 ± 2.71 a	2.93 ± 0.91 ab	24.51	9.01 ± 2.01 a	75.49
*Acacia–Hevea*	4.30 ± 1.20 a	8.58 ± 1.32 b	3.15 ± 1.37 a	36.69	5.43 ± 1.83 b	63.31

**Table 2 plants-13-00694-t002:** F-statistics from factorial a ANOVA assessing five plantation restoration types and litter layer (SL and UL) effects on the mass, total litter mass, and total litter thickness. df: degrees of freedom; SS: sum of squares; MS: mean square; ***: *p* < 0.001.

Variable	Factor	SS	df	F	MS	*p*
Litter mass	Forest type	106.027	4 80	26.507	7.99	<0.001 ***
	Litter layer	246.248	1 80	246.248	74.226	<0.001 ***
	Forest type × litter layer	47.01	4 80	11.752	3.543	0.010
Total mass	Forest type	212.015	4 40	53.004	5.302	0.002
Total thickness	Forest type	18.679	4 70	7.719	4.670	<0.001 ***

**Table 3 plants-13-00694-t003:** Soil bulk density, total porosity, capillary porosity, and non-capillary porosity for *A. mangium*, *H. brasiliensis*, *E. robusta*, *Acacia–Eucalyptus*, and *Acacia–Hevea*. Different lowercase letters indicate significant differences among the different plantation restoration types (*p* < 0.05).

Vegetation Type	Soil Depth (cm)	Bulk Density (g·cm^−3^)	Non-Capillary Porosity (%)	Capillary Porosity (%)	Total Porosity (%)
*A. mangium*	0 to 10	1.42 ± 0.07 b	1.95 ± 1.13 a	29.56 ± 7.25 a	31.51 ± 7.79 a
	10 to 20	1.56 ± 0.09 a	1.68 ± 0.85 a	30.84 ± 5.12 a	32.52 ± 5.04 a
	20 to 40	1.61 ± 0.08 a	1.53 ± 0.85 a	31.32 ± 2.89 a	32.86 ± 3.34 a
	40 to 60	1.62 ± 0.07 a	1.06 ± 0.59 a	31.24 ± 3.15 a	32.30 ± 3.19 a
*E. robusta*	0 to 10	1.52 ± 0.06 b	2.54 ± 3.14 a	27.73 ± 7.87 a	30.27 ± 6.13 a
	10 to 20	1.60 ± 0.05 a	2.68 ± 3.66 a	29.94 ± 3.07 a	32.62 ± 5.07 a
	20 to 40	1.63 ± 0.07 a	1.26 ± 0.85 a	31.00 ± 3.01 a	32.26 ± 3.19 a
	40 to 60	1.62 ± 0.08 a	0.94 ± 0.50 a	30.97 ± 4.53 a	31.91 ± 4.49 a
*H. brasiliensis*	0 to 10	1.45 ± 0.07 b	2.76 ± 1.83 a	31.36 ± 3.82 a	34.12 ± 5.13 a
	10 to 20	1.53 ± 0.06 a	2.38 ± 1.37 ab	32.59 ± 3.49 a	34.97 ± 3.31 a
	20 to 40	1.57 ± 0.06 a	1.77 ± 0.79 ab	33.31 ± 2.09 a	35.08 ± 2.04 a
	40 to 60	1.56 ± 0.10 a	1.23 ± 0.58 b	34.11 ± 1.92 a	35.35 ± 2.09 a
*Acacia–Eucalyptus*	0 to 10	1.41 ± 0.09 b	1.66 ± 1.05 a	30.62 ± 6.42 a	32.28 ± 6.89 a
	10 to 20	1.53 ± 0.07 a	1.74 ± 0.80 a	33.24 ± 3.55 a	34.98 ± 3.24 a
	20 to 40	1.57 ± 0.05 a	1.59 ± 1.31 a	33.11 ± 2.42 a	34.70 ± 2.58 a
	40 to 60	1.56 ± 0.06 a	1.23 ± 0.60 a	34.62 ± 2.34 a	35.85 ± 2.84 a
*Acacia–Hevea*	0 to 10	1.43 ± 0.04 b	2.29 ± 1.52 a	33.19 ± 3.13 a	35.47 ± 3.36 a
	10 to 20	1.53 ± 0.07 a	1.62 ± 1.57 a	33.51 ± 3.55 a	35.13 ± 4.08 a
	20 to 40	1.58 ± 0.09 a	1.38 ± 0.90 a	33.86 ± 2.26 a	35.24 ± 2.38 a
	40 to 60	1.52 ± 0.08 a	1.41 ± 0.81 a	35.23 ± 2.09 a	36.64 ± 2.19 a

**Table 4 plants-13-00694-t004:** Weighted values of water conservation capacity indexes in different plantation restoration types.

Grade I Index	Weight	Serial No.	Grade II Index	Weight
Litter layer	0.4429	P1	Litter thickness	0.0954
		P2	Total litter mass	0.0696
		P3	*W_m_*	0.1392
		P4	*Q_max_*	0.0697
		P5	*Q_eff_*	0.0690
Soil layer	0.5571	P6	Bulk density	0.0615
		P7	Non-capillary porosity	0.1560
		P8	Capillary porosity	0.0817
		P9	Total porosity	0.0912
		P10	Soil water holding capacity	0.1667

**Table 5 plants-13-00694-t005:** Comprehensive evaluation of water conservation capacity in different plantation restoration types.

Vegetation Type	Water Conservation Capacity		Comprehensive Evaluation Value	Rank
	Litter Layer	Soil Layer		
*A. mangium*	0.1097	0.1757	0.2854	1
*E. robusta*	0.1137	0.1363	0.2500	2
*H. brasiliensis*	0.0485	0.0520	0.1006	5
*Acacia–Eucalyptus*	0.0987	0.0753	0.1740	4
*Acacia–Hevea*	0.0722	0.1179	0.1900	3

**Table 6 plants-13-00694-t006:** Basic characteristics of the sampling sites.

Type of Community	Stand Age (Years)	DBH (cm)	Height (m)	Crown Width (m^2^)	Canopy Density (%)
*H. brasiliensis*	14	9.99 ± 1.45	10.49 ± 3.06	21.08 ± 17.66	86
*A. mangium*	14	13.28 ± 0.24	10.37 ± 3.50	44.55 ± 5.98	80
*E. robusta*	14	10.98 ± 1.96	12.08 ± 0.61	15.77 ± 12.46	65
*Acacia–Eucalyptus*	14	14.75 ± 3.57	11.73 ± 1.37	23.82 ± 4.04	80
*Acacia–Hevea*	14	14.23 ± 1.13	11.56 ± 0.64	40.18 ± 16.42	80

Note: DBH = diameter at breast height; data are presented as mean ± S.D.

## Data Availability

The data that support the findings of this study are available from the corresponding author upon reasonable request.

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
