# Peer review of "Differential Water Conservation Capacity in Broadleaved and Mixed Forest Restoration in Latosol Soil-Eroded Region, Hainan Province, China"

_plants, 2024, doi:10.3390/plants13050694_

Round 1

Reviewer 1 Report

Comments and Suggestions for Authors

Review comments for Plants manuscript 2840163

I reviewed the manuscript entitled “Differential Water Conservation Capacity in Broadleaved and Mixed Forest Restoration in Latosol Soil Eroded Region, Hai-nan province, China” authored by Suyi Chen, Yanping Huang, Mei Yan, Yujie Han, Kang Wang, Zexian Chen, Dongshuo Ruan, Yan Yu and Zhihua Tu.

This manuscript reports results of research evaluating water holding capacity of 5 forest restoration types planted on eroded soils in Hainan Province, China.

The Abstract is well composed and informative.

The Introduction is adequate in that it describes the problem and reason for the research. Several sentences are confusing and should be edited, for example:

Line 44 – for the sentence segment “41.10% was water erosion” should the word “was” be changed to “has?”

Line 44 – Should the word “areal” be changed to “area?”

Line 45 – the word “are” in “… China are water erosion” reads better if “are” is changed to “has.”

Several other sentences in the Introduction (particularly the last from Lines 85-90) would benefit from minor editing.

The Results section is well presented with adequate data provided in figures and tables. I found the figures informative and well prepared.

Line 92 – 3.1. Litter should be 2.1. Litter …

Line 183 – the text “nearly no changed” would read clearer if edited to “nearly unchanged.”

Discussion section

This section is adequate in that it describes with more details meaning of findings reported in the results section. Comparison of results from this study with results from other studies are provided.

Methods section

Line 405 – I believe the word “quadrate” is misspelled and should be “quadrat” – as meaning a small area where samples are collected.

Line 405 – Text is not clear about replications. The text reads as one location of each forest restoration type was selected and three sample sites established at that location, which seems to be subsamples rather than replication. This seems to be a situation known as pseudoreplication. True replication and stronger study design would be selecting at least two different planted tree locations of each species in the larger study area and collecting samples from each location. Replications should be better described. 

Line 420 – Is the DOI correct for reference [22]? I wanted to learn more about the indoor water soaking method as referenced in reference [22], but Google Scholar could not find this reference.

Line 458 – It would be helpful to state in section 4.5 (lines 455-458) the level of probability required for statistical significance, which is typically done for research papers of this type. Throughout the Results section, “p < .05” is stated for significant findings and the reader presumes this is the critical level for statistical tests.

Table 6 – The meaning of canopy width is not clear, and the unit (m2) implies an area measurement, not width. How was this measurement made; also how was canopy density measured? This could be included in as a footnote to the table, as done for DBH. Please include stand ages and basal areas if known in Table 6.

Conclusions section

This section is adequate, but minor editing appears necessary.

Line 463 – change “We study the …” to “We studied the …”

Line 464 – Consider changing the beginning of this sentence from “It found …” to “We found …”

Comments on the Quality of English Language

The overall quality of the English language is good in the manuscript. In my comments, however, I provided several suggestions where improvements should be considered. 

Author Response

Dear Reviewer:

Thank you for your comments concerning our manuscript entitled“Differential Water Conservation Capacity in Broadleaved and Mixed Forest Restoration in Latosol Soil Eroded Region, Hainan province, China”(ID: Plants-2840163). All comments are valuable and very helpful for revising and improving our paper, as well as the important guiding significance to our researches. We have studied comments carefully and have made correction which we hope meet with approval. Revised portion are using the "Track Changes" function in Microsoft Word. We explain point-by-point the details of the revisions in the manuscript and our responses to reviewers' comments. The main corrections in the paper and the responds to the comments are as flowing:

Point 1: Line 44 – for the sentence segment “41.10% was water erosion” should the word “was” be changed to “has?”

Response 1: Thank you for your comment. We have revised in Section of the sentence segment “41.10% was water erosion”. The word “was” had changed to “has”.

Point 2: Line 44 – Should the word “areal” be changed to “area?”

Response 2: Thank you very much. We have revised in line 44. The word “areal” had changed to “area”.

Point 3: the word “are” in “… China are water erosion” reads better if “are” is changed to “has.”

Response 3: Thank you very much. We have revised in line 45. The word “are” has changed to “has”.

Point 4: Several other sentences in the Introduction (particularly the last from Lines 85-90) would benefit from minor editing.

Response 4: Thank you very much. We have revised in line 85-90: “As a result, Eucalyptus forest was cut down to plant Acacia mangium forest instead. In addition, the restoration of vegetation has promoted changes in the structure of vegetation communities [19] and increased biodiversity, which affects the water-holding capacity of forests [20-26]. In order to understand the water-holding function after plantation restoration in the eroded area of Latosol, this study explores the differences between the litter and soil hydrological properties of five plantation restoration types in the Mahuangling Watershed, which is of great significance to provide a practical basis for plantation restoration in Latosol soil eroded region and improving the ecological environment.”

Point 5: Line 92 – 3.1. Litter should be 2.1. Litter …

Response 5: Thank you very much. We have revised in “2.1. Litter Thickness and Mass”.

Point 6: Line 183 – the text “nearly no changed” would read clearer if edited to “nearly unchanged.”

Response 6: Thank you very much. We have revised the text “nearly no changed” to “nearly unchanged”.

Point 7: Line 405 – I believe the word “quadrate” is misspelled and should be “quadrat” – as meaning a small area where samples are collected.

Response 7: Thank you very much. That is our misspel. We have revised the word “quadrate” to  “quadrat”.

Point 8: Line 405 – Text is not clear about replications. The text reads as one location of each forest restoration type was selected and three sample sites established at that location, which seems to be subsamples rather than replication. This seems to be a situation known as pseudoreplication. True replication and stronger study design would be selecting at least two different planted tree locations of each species in the larger study area and collecting samples from each location. Replications should be better described.

Response 8: Thank you very much. We selected three different planted tree locations of each species (each vegetation restoration type). We have revised the this section: “In July 2022, three 20 m × 20 m quadrats were established (this plot was selected did not cross the erosion trench and was at least 5 m away) in each vegetation restoration type to investigate the basic status of vegetation (Table 6).”

Point 9: Line 420 – Is the DOI correct for reference [22]? I wanted to learn more about the indoor water soaking method as referenced in reference [22], but Google Scholar could not find this reference.

Response 9: Thank you very much. The reference [22] is published in Chinese in the journal of “Research of Soil and Water Conservation”. Google Scholar could not find this reference, that can find this reference in the Homepages of this journal. Or you can find this reference for the link:  http://stbcyj.paperonce.org/oa/DArticle.aspx?type=view&id=20180414. Also, we can sending the PDF of the paper by email for you.

Point 10: Line 458 – It would be helpful to state in section 4.5 (lines 455-458) the level of probability required for statistical significance, which is typically done for research papers of this type. Throughout the Results section, “p < .05” is stated for significant findings and the reader presumes this is the critical level for statistical tests.

Response 10: Thank you very much. The ”p < .05” is stated for significant findings, we have revised in section 4.5: ”The significance level was set at p < 0.05. ”

Point 11: Table 6 – The meaning of canopy width is not clear, and the unit (m2) implies an area measurement, not width. How was this measurement made; also how was canopy density measured? This could be included in as a footnote to the table, as done for DBH. Please include stand ages and basal areas if known in Table 6.

Response 11: Thank you very much. In Table 6, the ”canopy width” should be “crown width”. In the plot, we measure the distance from east to west(m), and also measure the distance from north to south(m) for each tree. And the unit (m2) is the distance from east to west of the tree multiplied by the distance from north to south of the tree. 

And the canopy density refers to the ratio of the total projected area of forest canopy layer on the ground under direct sunlight to the total area of the stand area. We measured canopy density by the method of “line transects”, such as shown in the below picture: The projection length of the tree canopy on the two diagonals of the plot.

Point 12: Line 463 – change “We study the …” to “We studied the …”

Response 12: Thank you very much. We have changed “We study the …” to “We studied the …” in the text.

Point 13: Line 464 – Consider changing the beginning of this sentence from “It found …” to “We found …”

Response 13: Thank you very much. That is our misspel. We have revised “It found …” to “We found …”.

Special thanks to you for your good comments.

We tried our best to improve the manuscript and made some changes in the manuscript. These changes will not influence the content and framework of the paper.

We appreciate for your warm work earnestly, and hope that the correction will meet with approval.

Once again, thank you very much for your comments and suggestions.

Reviewer 2 Report

Comments and Suggestions for Authors

"Materia and methods" should be placed before the results.

Is the time of planting the same for all plantations?

Is the undergrowth situation the same in all plantations? Do differences in undergrowth affect differences in soil properties?

Please also include information on whether each tree species is evergreen or deciduous.

Table 5: Why do plantations combining acacia and eucalyptus rank low, despite the high rank of acacia and eucalyptus respectively? 

Author Response

Dear Reviewer:

Thank you for your comments concerning our manuscript entitled “Differential Water Conservation Capacity in Broadleaved and Mixed Forest Restoration in Latosol Soil Eroded Region, Hainan province, China”(ID: Plants-2840163). All comments are valuable and very helpful for revising and improving our paper, as well as the important guiding significance to our researches. We have studied comments carefully and have made correction which we hope meet with approval. Revised portion are using the "Track Changes" function in Microsoft Word. We explain point-by-point the details of the revisions in the manuscript and our responses to reviewers' comments. The main corrections in the paper and the responds to the comments are as flowing:

Point 1: "Materia and methods" should be placed before the results.

Response 1: Thank you very much for your constructive comments. In our manuscript, We use the Microsoft Word template of Plants to prepare our manuscript. In the template of Plants, research manuscript sections as follows: Introduction; Results; Discussion; Materials and Methods; Conclusions.

Point 2: Is the time of planting the same for all plantations?

Response 2: Thank you very much. In the 2000s, large-scale plantation restoration projects began to be implemented to improve the ecological environment in Mahuangling Watershed. And we selected the typical and representative vegetation restoration types in our study is planting in the 2008, which is planting the same time for all plantations.

Point 3: Is the undergrowth situation the same in all plantations? Do differences in undergrowth affect differences in soil properties?

Response 3: Thank you very much. In the plantations, the undergrowth situation is the same. The understory consists of species of Cyperus rotundus, Oplismenus undu-latifolius, Arthraxon hispidus, Embelia laeta, Schima superba, and Litsea glutinosa. However, the plant species numbers of understory are differences among in the five plantations. And the differences in undergrowth affect differences in soil properties, this is oue next step in this direction of research, especially, we will sutdy the effect on the soil chemical and microbiological properties. Thank you very much.

Point 4: Please also include information on whether each tree species is evergreen or deciduous.

Response 4: Thank you very much. Each tree species is evergreen. We have revised in section of “4.2. Litter and Soil Samples Collection”: The typical and representative vegetation restoration types we selected were Acacia mangium forest, Eucalyptus robusta forest, Hevea brasiliensis forest, Acacia-Eucalyptus forest (A. mangium and E. robusta), and Acacia-Hevea forest (A. mangium and H. brasiliensis), each tree species is evergreen.

Point 5: Table 5: Why do plantations combining acacia and eucalyptus rank low, despite the high rank of acacia and eucalyptus respectively?

Response 5: Thank you very much. Acacia-Eucalyptus forest (mix by A. mangium and E. robusta), that the construction of Mixed Forest is based on Eucalyptus forest was artificial regeneration by strip-felling and then plant Acacia mangium instead. The tree of Eucalyptus was logging by machinery, and that caused different degrees of soil compaction, which maybe caused the rank is low. The further studies are still needed. Thank you very much again.

Special thanks to you for your good comments.

We tried our best to improve the manuscript and made some changes in the manuscript. These changes will not influence the content and framework of the paper.

We appreciate for your warm work earnestly, and hope that the correction will meet with approval.

Once again, thank you very much for your comments and suggestions.
